# Effect and cost-effectiveness of human-centred design-based approaches to increase adolescent uptake of modern contraceptives in Nigeria, Ethiopia and Tanzania: Population-based, quasi-experimental studies

**Catarina Krug**[1]*, **Melissa Neuman**[1,2], **James E. Rosen**[3], **Michelle Weinberger**[3], **Stefanie Wallach**[4], **Mary Lagaay**[4], **Melanie Punton**[4], **Annapoorna Prakash**[1], **Mussa Kelvin Nsanya**[5], **Philip Ayieko**[1,2,5], **Saidi Kapiga**[1,5], **Yewande P. Ajayi**[6], **Emily E. Crawford**[6], **Eskindir Tenaw**[7], **Mohammed Mussa**[7], **Som Kumar Shrestha**[1], **Christian Bottomley**[1,2], **James R. Hargreaves**[8], **Aoife Margaret Doyle**[1,2]

1 Department of Infectious Disease Epidemiology, London School of Hygiene and Tropical Medicine, London, United Kingdom, 2 MRC International Statistics and Epidemiology Group, London, United Kingdom, 3 Avenir Health, Washington, DC, United States of America, 4 Itad, Hove, United Kingdom, 5 Mwanza Intervention Trials Unit, National Institute for Medical Research, Mwanza, Tanzania, 6 Binomial Optimus Limited, Abuja, Nigeria, 7 MMA Development Consultancy, Addis Ababa, Ethiopia, 8 Department of Public Health, Environment and Society Research, London School of Hygiene and Tropical Medicine, London, United Kingdom

* denoronhakrug@hotmail.com

## Abstract

Around half of adolescent pregnancies in low- and middle-income countries are unintended, contributing to millions of unsafe abortions per year. Adolescents 360 (A360), a girl-centred initiative, aimed to increase voluntary uptake of modern contraceptives among adolescents in Nigeria, Ethiopia and Tanzania. We evaluated the effectiveness and cost-effectiveness of A360 in increasing modern contraceptive use in selected geographies. We used before-and-after cross-sectional studies of adolescent girls in four settings. Two Nigerian settings had purposefully selected comparison areas. Baseline and endline household surveys were conducted. The primary study outcome was modern contraceptive prevalence rate (mCPR). Secondary outcomes mapped onto the A360 Theory of Change. Interpretation was aided by a process evaluation along with secular mCPR trends and self-reported A360 exposure data. Incremental design and implementation costs were calculated from implementer systems, site visits, surveys, and interviews. mCPR change was modelled into maternal disability-adjusted life years (DALY) averted to calculate incremental cost-effectiveness ratios. In Oromia, Ethiopia, mCPR increased by 5% points (95% CI 1–10; n = 1,697). In Nigeria, there was no evidence of an effect of A360 on mCPR in Nasarawa (risk ratio: 0·96, 95% CI: 0·76–1·21; n = 5,414) or in Ogun (risk ratio: 1·08, 95% CI: 0·92–1·26; n = 3,230). In Mwanza, Tanzania, mCPR decreased by 9% points (-17 to -0.3; n = 1,973). Incremental cost per DALY averted were $30,855 in Oromia, $111,416 in Nasarawa, $30,114 in Ogun, and $25,579 in Mwanza. Costs per DALY averted were 14–53 times gross domestic

**Data Availability Statement:** Data is available at the LSHTM Data Compass: https://doi.org/10.17037/DATA.00003599.

**Funding:** This work was supported by the Bill & Melinda Gates Foundation [OPP1134172] (SW, ML, AD, MN, MP) and the Children's Investment Fund Foundation (SW, ML, AD, MN, MP). The content is solely the responsibility of the authors and does not necessarily represent the official views of the funders.

**Competing interests:** The authors have declared that no competing interests exist.

product per capita. A360 did not lead to increased adolescent use of modern contraceptives at a population level, except in Oromia, and was not cost-effective. This novel adolescent-centred design approach showed some promise in addressing the reproductive health needs of adolescents, but must be accompanied by efforts to address the contextual drivers of low modern contraceptive use.

## Introduction

Around half of adolescent pregnancies in low- and middle-income countries are unintended [1]. Ensuring that adolescent girls have access to sexual and reproductive health-care services is critical for achieving universal access (Sustainable Development Goal 3.7). Moreover, being able to control their fertility underpins educational and employment opportunities [2]. Although many programs have sought to increase contraceptive use among adolescents, their effectiveness has been limited [3, 4].

Adolescents 360 (A360) was a four-year (2016–2020) initiative to increase voluntary uptake of modern contraceptives among adolescent girls in four settings. A360 used human-centred design (HCD) to develop setting-specific interventions through an iterative process of research and prototyping. The program hypothesis was that meaningful engagement of adolescents would catalyse the development of novel, successful approaches. The expectation was that higher design costs would be offset by better-designed and more effective interventions to produce a cost-effective approach.

We describe outcome evaluation and cost-effectiveness studies of A360 in Northern Nigeria, Southern Nigeria, Ethiopia and Tanzania, and draw on process evaluation findings to interpret the results.

## Methods

### Study design

The study design is described in detail elsewhere [5]. We used repeat cross-sectional surveys to evaluate the impact of A360 on modern contraceptive use. In Nigeria, the study design included a comparison area purposively selected by the implementers, Society for Family Health, in collaboration with the state Ministry of Health and local government officials. Comparison-intervention pairs (two in Northern Nigeria and one in Southern Nigeria) were selected to be similar with respect to some or all of the following criteria: population density, estimated modern contraceptive prevalence rate (mCPR) among 15 to 49 year olds, number of health facilities and presence of World Bank support for Maternal and Child Health activities. In each setting, eligible girls were identified at the household level. The smallest available administrative unit was used as the primary sampling unit–enumeration area in Nigeria, kebele in Ethiopia and street in Tanzania. Study areas were selected in collaboration with the implementers. The same study design and the same primary sampling units were included at baseline and endline, where possible. Although the design means that it is possible that in each site the same households and individuals may be included in the baseline and endline surveys, no attempt was made to trace individuals or households from baseline to endline.

The outcome and cost-effectiveness studies were conducted in the same locations. In Northern Nigeria, 'Matasa Matan Arewa' ('Adolescent Girls from the North') was evaluated in four local government areas in Nasarawa state; in Southern Nigeria, '9ja Girls' was evaluated in

two local government areas in Ogun state. In Ethiopia, 'Smart Start' was evaluated in four woredas (districts) in Oromia state. In Tanzania, 'Kuwa Mjanja' ('Be Smart') was evaluated in Ilemela district in Mwanza region.

## Participants

We included girls aged 15 to 19 years who would be eligible for the A360 intervention. In Nasarawa and Oromia, we included girls who were married or living as married; in Ogun, we included girls who were unmarried; and in Mwanza, we included girls who were married or unmarried.

In Nasarawa and Ogun, we collected baseline data between August and September 2017 and endline survey data between November and December 2020. In Oromia, we collected baseline data between September and October 2017 and endline data between November and December 2020. In Mwanza, we collected baseline data between September 2017 and January 2018 and endline data between May and October 2021.

Individual, written consent, was obtained from all participants before conducting the interviews. In Ogun, parental/guardian consent and adolescent girl assent were required for unmarried girls aged up to 17 years. In Mwanza, parental consent waiver was granted for this age group because of the sensitive nature of the survey.

## Sample size

The target sample size of 23,481 (Nasarawa 4,555, Ogun 12,020, Oromia 1,926, Mwanza 4,980) was chosen to provide 90% power to detect an intervention effect at each of the study sites. Sample size calculations took into account the design effect (clustering), estimated non-response, and the fact that not all girls were sexually active. We assumed A360 would increase mCPR from: 3·0% to 5·1% in Nasarawa, 64·4% to 72·6% in Ogun, 44·0% to 50·8% in Oromia and 26·7% to 32·7% in Mwanza. The effect estimates and baseline mCPR rates were derived from a review of 25 studies as detailed in the protocol paper [5]. In Nasarawa, Ogun and Mwanza, baseline mCPR was higher than expected, which led to revised endline sample sizes (S1 Text).

## Interventions

The final package of interventions was site-specific and is presented in Table 1, according to template for intervention description and replication (TIDieR) checklist and guide [6]. The A360 program tracked other sexual and reproductive health interventions in intervention (and comparison areas in Nigeria), which are also summarized in Table 1 [7].

## Objectives

Our primary aim was to evaluate the effectiveness and cost-effectiveness of the A360 approach in increasing modern contraceptive use among sexually active girls aged 15–19 years. Our secondary aims align with the A360 Theory of Change components and are described in S1 Table. We also quantified the association between the respondents' self-reported exposure to A360 and primary and secondary outcomes.

## Outcomes

The primary study outcome was the proportion of fecund and sexually active girls who reported using modern contraception at the time of the surveys (mCPR). Modern contraception included male and female sterilisation, contraceptive implants, intrauterine contraceptive

**Table 1. Description of Adolescents 360 program components by site.**

| Item no. | Item | Nasarawa, Nigeria | Ogun, Nigeria | Oromia, Ethiopia | Mwanza, Tanzania |
|---|---|---|---|---|---|
| 1 | **Intervention name.** | Matasa Matan Arewa (MMA) | 9ja Girls | Smart Start | Kuwa Mjanja |
| 2 | **Geographies where it occurred.** | Seven selected LGAs in two states: Nasarawa (Doma and Karu LGAs) and Kaduna (Chikun, Igabi, Sabon gari, and Zaria LGAs) [8] | 19 LGAs in seven states in Southern Nigeria: Ogun (Abeokuta south and Ado-Odo/Ota LGAs), Lagos (Agege and Alimosho LGAs), Osun (Iwo LGA), Oyo (Akinyele and Ibadan north-east LGAs), Edo (Ikpoba Okha and Oredo LGAs), Akwa Ibom, and Delta (Warri south LGA) [9] It was also implemented in some areas of Kaduna state, in Northern Nigeria. | 39 Woredas (districts) in four regions: Amhara, Oromia, Southern Nations Nationalities and Peoples, and Tigray [10] | 20 Regions: Arusha, Dar es salam, Dodoma, Geita, Iringa, Kagera, Katavi, Kilimanjaro, Lindi, Manyara, Mara, Mbeya, Morogoro, Mtwara, Mwanza, Pwani, Rukwa, Ruvuma, Shinyanga, Simiyu, Songwe, Tabora, and Tanga. |
| 3 | **Timing of intervention.** | In Karu LGA: from April 2018 in 2% (n = 5/261)[1] of all health facilities in the LGA; the intervention program was delivered over a period of 31 months in total. In Doma LGA: from June 2019 in 9% (n = 5/54)[1] of all health facilities in the LGA; for 18 months. | In Ado-Odo/Ota: from December 2017 in 9% (n = 13/147)[1] of all health facilities in the LGA; for 36 months. | In Fentale woreda: from April 2018; the intervention program was delivered in different kebeles within the woreda over a period of 31 months in total. In Ada'a woreda: from June 2018; for 29 months. In Lome and Wara Jarso woredas: from August 2018; for 27 months. [2] Implementation was conducted in a staggered way across kebeles and by the end of 2020, all kebeles were expected to have receive the intervention.[3,4] | In Mwanza, the intervention program was delivered from January 2018 to September 2020; for 32 months. |
| 4 | **Reach by end of September 2020** | 45,371 adolescent girls | 172,517 adolescent girls | 75,237 adolescent girls | 314,155 adolescent girls |
| 5 | **Rationale, theory, or goal of the elements essential to the intervention.** | Establishes the relevance of contraception for married adolescent girls and their husbands by linking birth spacing to family health and girls' life goals. Female mentors deliver life and vocational skill sessions to girls, and male mobilizers start conversations with husbands to encourage referrals to walk-in counselling | Combines walk-in contraceptive counselling with life-skill sessions for unmarried girls. Uses life and vocational skills as an entry point to engage unmarried adolescent girls in conversations about contraception and how it can be a tool to help them achieve their life goals. | Engages married adolescent girls and their husbands, using financial planning as an entry point to discuss contraception and help them understand how delayed first birth and spaced pregnancies facilitate improved savings and capital and financial security to pursue their shared life goals. | Engages mainly unmarried adolescent girls around their life aspirations. Provides them with low-intensity vocational skills sessions and contraceptive service delivery. Aims to give them the tools they need to balance their growing responsibility and navigate the social transition to adulthood. |
| 6 | **Physical or informational materials used in the interventions, including those provided to participants or used in intervention delivery or in training of intervention providers.** | (a) Life, Family and Health (LFH)—Four mentored group sessions whose curriculum focuses on nutrition, sexual and reproductive health, life skills and vocational skills. (b) Contraceptive counselling through an 'opt-out' moment at the end of each LFH session or to walk-in clients. | (a) Life, Love and Health (LLH)—One facilitated group session whose curriculum features vocational skills, future-planning exercises, and discussions about love, sex and dating. (b) Contraceptive counselling through an 'opt-out' moment at the end of each LLH session or to walk-in clients. | Girls engage in a Smart Start session guided by: (a) Discussion aide which introduces concepts around financial planning through easy to understand visuals and links these concepts to delayed first birth and spaced pregnancies. (b) Goal card it was used for married adolescent girls to record the aspirations they discuss in the counselling session and use as a conversation starter with their husbands or family members. (c) Contraceptive counselling through an 'opt-out' moment at the health post. | (a) Single group session implemented as part of in and out-of-clinic events. During the sessions, girls participate in a life and vocational skills induction session where they learn an entrepreneurship skill from a trained provider–e.g. jewellery or soap making. (b) 'Kuwa Mjanja Queens' use interactive games about contraceptive choices and side effects, with the help of tablets containing the 'Mjanja Connect' app, to engage girls between activities. (c) Contraceptive counselling through an 'opt-out' counselling moment at each event. |

*(Continued)*

**Table 1.** (Continued)

| Item no. | Item | Nasarawa, Nigeria | Ogun, Nigeria | Oromia, Ethiopia | Mwanza, Tanzania |
|---|---|---|---|---|---|
| 7 | Description of each of the procedures, activities, and/or processes used in the intervention. | (a) Female mentors conduct door-to-door recruitment of girls into cohorts for LFH sessions (b) Male Interpersonal Communicators start conversations with groups of men in public spaces or traditional joints. Mobilizers use the health of the baby and mother to encourage married men to refer their adolescent wives to a clinic for counselling. (c) Girls referred by their husbands then attend a health clinic for a one-to-one appointment with a provider [8] | (a) Girls are reached and recruited for LLH sessions or signposted to walk-in appointments through community mobilizers, their friends—through peer to peer referrals—or their mothers. (b) Health providers engage mothers of adolescent girls through two sessions. The aim of these is to help mothers understand that contraception can be a tool to help girls achieve their dreams. These sessions also aim to dispel myths and misconceptions around contraception. (c) Girls LLH sessions or attend a health clinic for a one-to-one appointment with a provider [9] | (a) Members of the Women's Development Army, Health Extension Workers, Youth Champions (peer volunteers) or A360-employed Smart Start Navigators mobilize girls and their couples through door-to-door visits. Health Extension Workers conduct financial and contraceptive counselling. | (a) Girls are mobilized through public announcements delivered by A360 personnel or community mobilizers, through school-based mobilization, through peers ('Kuwa Mjanja Queens') who visit girls in their homes, and/or through their parents or friends. (b) Parents are invited to attend a single session to start conversations about contraception and encourage participants to support their daughters to attend events. (c) Community engagement: Early and frequent engagement with government officials, advocacy meetings and sharing data and results. |
| 8 | For each category of intervention provider, description of their expertise, background and any specific training given. | (a) Female mentors who moderate LFH sessions are trained by the A360 approach. (b) Contraceptive counselling is delivered by A360 Young Providers, who are recruited by A360, and work full time in public health clinics. It may also be delivered by government health workers. All providers are trained in youth-friendly service provision and use counselling protocols that focus on issues that are of most concern to girls. | Same as for MMA | The Session is delivered by the Smart Start Navigators or Health Extension Workers, who are trained to host conversations about financial planning and provide contraceptive services in an approachable way to rural, married girls and their husbands. | (a) 'Kuwa Mjanja Queens' are trained by A360 on reaching their peers. (b) Community Development Officers or Youth Development Officers (a government personnel at the district level) coordinate and provide entrepreneurship skills during events. (c) Contraceptive counselling and services are delivered by government health care workers. They receive a short orientation on Kuwa Mjanja but no direct training |
| 9 | Description of delivery models (e.g. face-to-face) of the intervention and whether it was provided individually or in a group. | (a) LFH sessions are conducted through face-to-face sessions, in groups of 12. (b) Contraceptive counselling is provided individually after each session if a girl does not 'opt-out'. (c) Girls can also access walk-in appointments with trained providers at any time at their nearest clinic | (a) LLH sessions are conducted face-to-face; provided in groups of 15–20 girls. (b) Contraceptive counselling is provided individually after each session if a girl does not 'opt-out'. (c) Girls can also access walk-in appointments with trained providers at any time at their nearest clinic (d) Mothers' sessions are conducted in groups. | The session is conducted face-to-face, to the girl (individually), or to the couple. | (a) Sessions are implemented face-to-face and events are conducted in groups. (b) Contraceptive counselling is provided individually unless a girl wishes to 'opt-out' |
| 10 | Description of the type(s) of location(s) where the intervention occurred, including any necessary infrastructure or relevant features. | (a) The two first LFH classes happen at the traditional leader's house, in the community, or at the primary health centre. The 2nd and 4th sessions happen at the primary health centre. (b) Contraceptive counselling is offered at service delivery points at health facilities. (c) Male Interpersonal Communicators reach husbands of adolescent girls in private outdoor spaces such as traditional joints and places where men gather to relax. Interested husbands are given referral cards for their wives to either join a LFH class or visit a provider at the facility. | (a) After hearing about 9ja Girls, the girl can drop in to a LLH class at primary healthcare clinics or choose to go directly to a nearby public healthcare clinic for a walk-in appointment. | (a) The session is conducted at the girl's household, in convenient locations at the community, or at the health post. (b) Contraceptive counselling is offered at the health post. (c) Community engagement sessions are conducted at the community. | Kuwa Mjanja events happen either at a health facility (in clinic event) or in pop-up tents within the community spaces or other community facilities (out of clinic event). |

*(Continued)*

**Table 1.** (Continued)

| Item no. | Item | Nasarawa, Nigeria | Ogun, Nigeria | Oromia, Ethiopia | Mwanza, Tanzania |
|---|---|---|---|---|---|
| 11 | **Description of the number of times the intervention was delivered and over what period of time including the number of sessions, their schedule, and their duration, intensity or dose.** | Mentored sessions consist of four sessions conducted every two weeks. Walk-in appointments are available at any time the clinic is open. | One single LLH session is conducted for each group of girls, on Saturdays. Walk in appointments are available at any time the clinic is open | One single Smart Start session is conducted for each girl or couple. After an initial 6-week implementation period, Smart Start Navigators move on to a different community, leaving Health Extension Workers and Women's Development Army to continue implementing the program with the support of regional A360 and government staff. | Kuwa Mjanja session is delivered once per community. Outreach teams work on a rotating schedule: out-of-clinic and in-clinic events are implemented in one district in one month and again approximately three months later. |
| 12 | **If the intervention was planned to be personalised, titrated or adapted, then describe what, why, when, and how.** | The intervention is subjected to continuous quality programme improvement through adaptive implementation. This involves the use of routine mixed methods data to identify real time areas for adaptation for improved effectiveness. | Same as for MMA | Same as for MMA | Same as for MMA |
| 13 | **If the intervention was modified during the course of the study, describe the changes (what, why, when, and how).** | Some of the intervention adaptations are captured in the process evaluation reports; others are captured in program reports and adaptation audits [7]. | Same as for MMA | Same as for MMA | Same as for MMA |
| 14 | **Description of how and by whom intervention adherence or fidelity was assessed.** | Use of insights from the process evaluation, client-exit interviews, program monitoring metrics and periodic quality of care assessments. | Same as for MMA | Same as for MMA | Same as for MMA |
| 15 | **Actual: If intervention adherence or fidelity was assessed, describe the extent to which the intervention was delivered as planned.** | These can be found in the process evaluation reports and the program reports submitted to the donors. | Same as for MMA | Same as for MMA | Same as for MMA |

(*Continued*)

**Table 1.** (Continued)

| Item no. | Item | Nasarawa, Nigeria | Ogun, Nigeria | Oromia, Ethiopia | Mwanza, Tanzania |
|---|---|---|---|---|---|
| 16 | **Similar interventions in study areas** | In Doma LGA, Nasarawa State, there were no further interventions in place besides MMA. In Karu LGA, also in Nasarawa State, Marie Stopes International [11] and Planned Parenthood Federation of Nigeria [12] were conducting outreach activities on family planning counseling and services for White Ribbon Alliance [13]. [4] | Ado-Odo/Ota LGA, Ogun State, had The Challenge Initiative intervention in place since 2018. The intervention implemented community demand generation of adolescent sexual and reproductive health services especially with their 'Life Planning Ambassadors' targeting adolescent girls/youths aged 15–24 years. | Oromia Development Association in association with The David and Lucile Packard Foundation, developed a sexual and reproductive health program aimed at reducing early marriage in the Oromia region of Ethiopia. It was a school-based program delivered by trained teachers [14]. In Fentale woreda, Oromia Development Association targeted girls aged 15–19 years. It used Behavior Change Communication and Information, Education and Communication. The program also included community engagement, referral linkage to facilitate provision of contraceptive methods at health centers, as well as training of staff from health centers for youth-friendly sexual and reproductive health services [14]. In Lome woreda, Oromia Development Association developed school-based learning activities to empower girls and boys aged 12–15 years (primary; grade 5 to grade 8) with relevant sexual and reproductive health awareness [15]. The program included game-based learning activities focused on menstrual hygiene and management concerns of schoolgirls and it was in place since 2009 [15, 16]. In Wara Jarso and in Ada'a woredas, we are not aware of any additional interventions which were targeting adolescent girls besides A360. | In addition to A360, in Mwanza region, the Marie Stopes Tanzania, a part of Marie Stopes International, provided contractive and sexual health services. Marie Stopes Tanzania provided a range of contraceptives according to the clients' needs along with the provision of sexual and reproductive health counselling. In Mwanza, Marie Stopes Tanzania, has set clinics were people can get information about their reproductive choices and access high quality family planning services. Another such program was the Beyond Bias initiative in Tanzania. Beyond Bias was an initiative funded by the Bill & Melinda Gates Foundation that is working towards ensuring judgement-free and quality counselling along with the provision of contractive to young people in Tanzania. Led by Pathfinder International, in Tanzania, Pathfinder is working with the Ministries of Health. Beyond bias aims to identify different types of healthcare provider biases that compromise the successful dissipation of family planning services. With the human-centred design, these solutions to address these biases are identified to ensure equitable and judgement-free access to contraceptives. |

[1] Source of information: Population Services International (PSI), 03/03/2021.

[2] Start dates were defined using PSI reports and monitoring data. PSI revised and confirmed these start dates with the on 09/02/2021.

[3] 'Specifically, all kebeles in Fentale received the interventions by May 2019, in Ada'a by January 2020, in Lome by December 2020 and in Wara Jarso by May 2020'.

[4] Source of information: Document entitled "A360 OE site mapping" shared by PSI, on 12/11/2020.

Note: PSI led the overall design and implementation of Adolescents 360 and worked in partnership with IDEO.org and the Center for the Developing Adolescent at the University of California at Berkeley. Country work was led by PSI Ethiopia, PSI Tanzania, and Society for Family Health, a non-governmental organisation in Nigeria. Implementation was accompanied by a monitoring and evaluation component to examine process level indicators such as the number of girls reached. MMA, Matasa Matan Arewa, LGA, local government area, LFH, Life Family and Health, LLH, Life Love and Health, A360, Adolescents 360 intervention

devices, injectables, contraceptive pill/oral contraceptives, emergency contraceptive pill, male condom, female condom, Standard Days Method, Lactational Amenorrhoea Method, diaphragm, spermicides, foams and jelly [5]. To better understand the pathways through which the A360 approach could affect mCPR, secondary outcomes were also measured (S1 Table).

One of the secondary outcomes was proportion of current modern contraceptive users who were using a long-acting reversible contraceptive (LARC), which was measured due to the large global emphasis put on this type of methods to offer girls with access to the widest available contraceptive options.

Questionnaires were adapted from Demographic and Health Survey and Family Planning 2020 survey instruments and were pre-tested for comprehension, flow, appropriateness and feasibility of implementation. At baseline, all questionnaires were administered face-to-face, whereas at endline, in Nigeria and Ethiopia, the first part of the questionnaire was administered face-to-face while the second part of the questionnaire was administered by phone due to COVID-19 related restrictions on data collection (S1 Text).

## Statistical methods

The impact of A360 interventions was assessed by quantifying change between baseline and endline, guided by a pre-specified analysis plan (S1 Text). In Nasarawa and Ogun, we used a difference in difference approach using Poisson regression (binary outcomes) or linear regression (continuous outcomes). The model included area (intervention versus comparison), time, and an interaction between area and time. The interaction term coefficient, reflecting the effect of A360 beyond the time trend, was used to assess impact. Models additionally included age, education level, number of living children, religion and wealth quintile. Clustering at the level of the primary sampling unit was accounted for by using cluster-robust standard errors to calculate p-values and confidence intervals. Differences between intervention and comparison areas at baseline were tested using t-test or Pearson $\chi^2$ test, as appropriate. The validity of the difference in difference approach depends on the assumption that, if the intervention had not occurred, the trend in mCPR would have been the same in the intervention and comparison areas [17]. To assess these trends, we used Health Management Information System service data.

In Oromia and Mwanza, linear regression models were fitted to data from the baseline and endline surveys aggregated at the level of the primary sampling unit. Matching was accounted for by including the primary sampling unit in the regression model as a categorical variable. The models also included time and the confounders variables listed above. Observed changes in mCPR could be due to secular trends [5]. Therefore, we examined trends over time in study settings, using secondary datasets. The details of this analysis are presented in S1 Text.

The association between self-reported exposure to A360 and mCPR was evaluated using data from intervention areas at endline. A girl was considered exposed if she reported hearing about A360 interventions available in the place where she lived (S1 Text). Poisson (Nasarawa and Ogun) or logistic regression (Oromia and Mwanza) were used for binary outcomes and linear regression was used for continuous outcomes. The models included a variable denoting exposure status, and the confounder variables listed above. Cluster-robust standard errors were used to account for the clustering at the primary sampling unit level.

## Costing and cost-effectiveness

The cost-effectiveness analysis defined the comparator for A360 as the status quo for design and implementation of adolescent programming. The design comparator was Population Services International's (PSI) DELTA design methodology [18], the standard used at the time A360 initiated. The implementation comparator was the existing contraceptive programming available to adolescents in the A360 study geographies. We estimated incremental cost-effectiveness ratios for each study setting. Incremental costs were A360 design and implementation costs minus the comparator cost. See S1 Text for additional details.

### Role of the funding source

The final study designs were discussed and agreed upon with the funders. The funders had no role in the data collection, data analysis, or writing of the manuscript. The corresponding author had full access to all the data in the study and had final responsibility for the decision to submit for publication.

### Ethics approvals

The study components were approved by the National Health Research Ethics Committee of Nigeria (NHREC/01/01/2007-25/05/2017; NHREC/01/01/2007-23/06/2020C; NHREC/01/01/2007-15 /03/2019C), the Oromia Health Bureau Research Ethical Review Committee in Ethiopia (BEFOIHBTFH/1-8/2844; BEF0/AHBTFH/1-16/3089), the Addis Ababa University, College of Health Sciences Institutional Review Board (074/16/SPH), the National Health Research Ethics Review sub-Committee of Tanzania (NIMR/HQ/R.8a/Vol. IX/2549; NIMR/HQ/R.8a/Vol. IX/3508; NIMR/HQ/R.8a/Vol.IX/2346) and the London School of Hygiene and Tropical Medicine Ethics Committee (Ref: 14145).

## Results

### A360 outcome evaluation

Survey response rates were between 69% and 100% (Fig 1). Selected sociodemographic characteristics of girls in the baseline sample are presented in Table 2. S2 Table illustrates differences in number of children, religion, wealth and mobile phone access in intervention and comparison areas at baseline.

### Nasarawa, Nigeria

In Nasarawa, mCPR increased from 16% to 38% in the intervention areas and from 13% to 27% in comparison areas. After accounting for the change in mCPR in comparison areas via the difference in differences analysis, we found no evidence of an impact of A360 (risk ratio: 0·96; 95% CI: 0·76–1·21; p = 0·734; Table 3).

We observed intervention effects in two out of 17 secondary outcomes. For instance, girls in intervention areas reported a greater increase over time in positive attitudes to couples using modern contraceptives compared to those in comparison areas (S3 Table).

Self-reported exposure to 'Matasa Matan Arewa' was low in intervention areas, varying between 5% (95% CI: 4–6) and 7% (95% CI: 5–10). There was a positive association between self-reported exposure and mCPR (risk ratio 1·41; 95% CI: 1·13–1·76; p<0·001; Table 2), and between self-reported exposure and 10 out of 17 secondary outcomes (S4 Table).

### Ogun, Nigeria

In Ogun, mCPR increased from 45% (95% CI: 41–48) to 49% (95% CI: 44–53) in intervention areas but remained constant at 51% in comparison areas (Table 2). In the difference and differences analysis, we found no evidence of an impact of A360 (risk ratio, 95% CI: 1·08, 0·92–1·26; p = 0·340; Table 3).

Out of 17 secondary outcomes, the only effect observed in the hypothesised direction was on the proportion of LARC users among all modern contraceptive users which increased from 0·3% to 2·2% at the intervention area and dropped from 1·4% to 1·0% at the comparison area (S3 Table).

Self-reported exposure to 9ja Girls was 8% (95% CI: 6–10) in the intervention area. There was no relationship between self-reported exposure and primary and secondary outcomes,

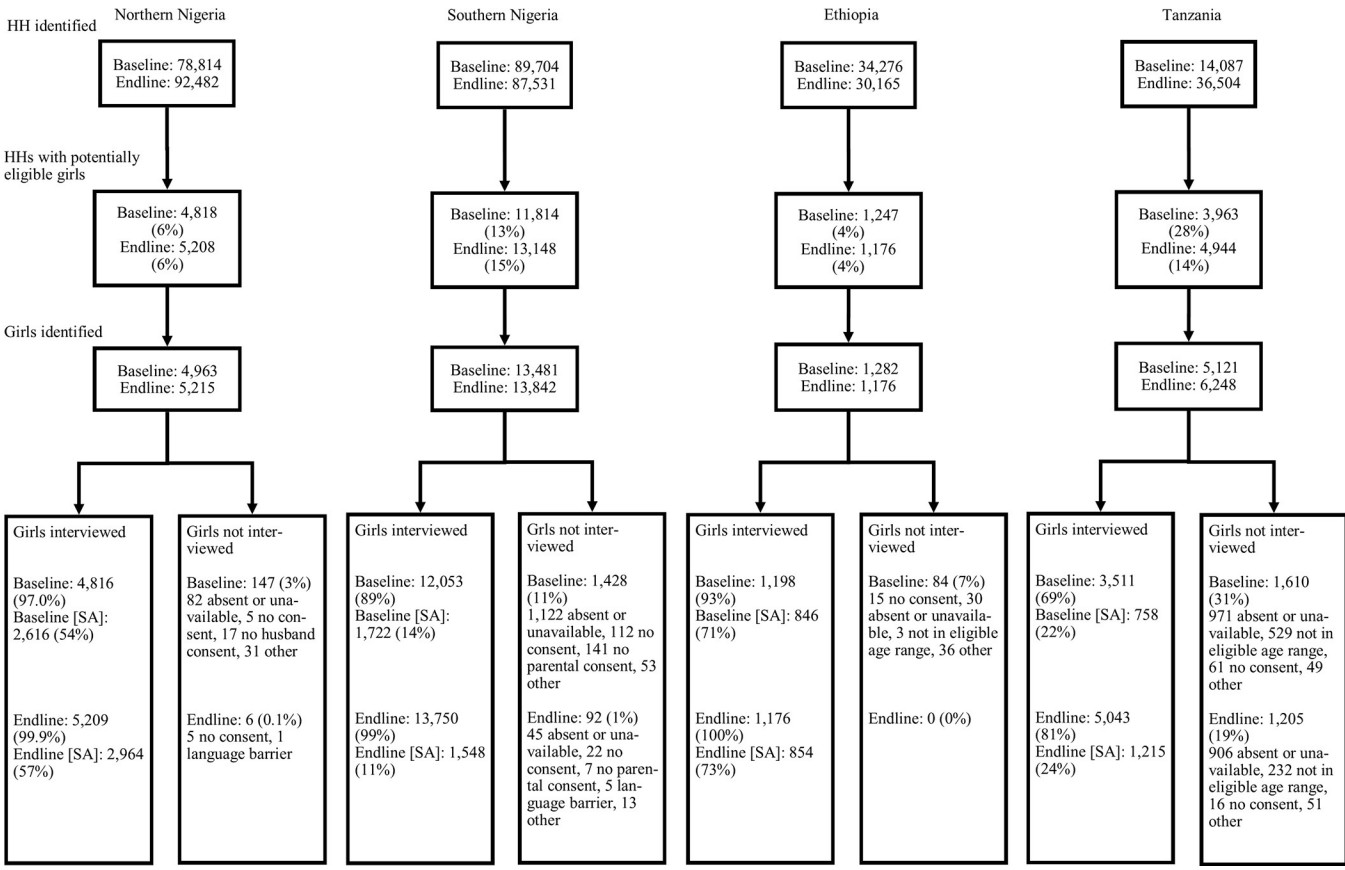

**Fig 1. Flow diagram.** HH, households, SA, fecund and sexually active girls during the 12 months before the interview.

except for a weak evidence of a positive relationship between being exposed and being aware of contraceptive products (S4 Table).

## Oromia, Ethiopia

In Oromia, the kebele-average mCPR was 64% (95% CI: 57–71) at baseline and 68% (95% CI: 62–75) at endline. In the adjusted analysis, we estimated that mCPR increased by five percentage points between baseline and endline surveys (95% CI: 1–10; p = 0·025; Table 3).

This increase in mCPR was accompanied by an increase in four out of 17 secondary outcomes (S4 Table). For instance, there was a 10% absolute increase in the proportion of current modern contraceptive users using a LARC (95% CI: 3–17; p = 0·004), and an 11% increase (95% CI: 1–21; p = 0·025) in awareness of contraceptive products. There was also evidence of an in increase in the proportion of adolescent with positive attitudes towards the use of modern contraceptives (S3 Table).

Self-reported exposure to Smart Start was 24% (95% CI: 18–30). There was a positive association between self-reported exposure and mCPR (odds ratio: 2·09; 95% CI: 1·32–3·29; p = 0·002; Table 3), and between self-reported exposure and two out of 17 secondary outcomes (S4 Table).

## Mwanza, Tanzania

The eligible population of girls was primarily unmarried sexually active girls (around 94%). mCPR was 50% (95% CI: 46–54) at baseline and 40% (95% CI: 37–44) at endline. Following

**Table 2. Demographic characteristics of adolescent girls in intervention areas at baseline[a].**

| Demographic characteristics | Levels | Nasarawa, Nigeria | Ogun, Nigeria | Oromia, Ethiopia | Mwanza, Tanzania |
|---|---|---|---|---|---|
| **Age (years)** | | 17·54 (0·04) | 16·88 (0·02) | 17·80 (0·06) | 16·91 (0·02) |
| **Number of living children** | No children | 1,245 (53%) | 5,903 (98%) | 600 (50%) | 3,535 (92%) |
| | 1 child | 755 (32%) | 114 (2%) | 520 (44%) | 246 (7%) |
| | 2 children | 296 (13%) | 4 (0%) | 74 (6%) | 29 (1%) |
| | 3 or more children | 46 (2%) | 1 (0%) | 4 (0%) | 1 (0%) |
| **Education level** | No education | 642 (27%) | 81 (1%) | 327 (31%) | 130 (4%) |
| | Qur'anic only | 61 (3%) | 1 (0%) | N/A | N/A |
| | Primary | 548 (23%) | 263 (4%) | 695 (55%) | 1,339 (38%) |
| | Secondary | 1,039 (44%) | 5,407 (90%) | 172 (14%) | 1,935 (55%) |
| | Higher/Technical | 51 (2%) | 270 (4%) | 4 (0%) | 107 (3%) |
| | Don't know | 1 (0%) | 0 (0%) | 0 (0%) | 0 (0%) |
| **Religion** | Roman Catholic | 406 (17%) | 125 (2%) | 0 (0%) | 1,376 (39%) |
| | Orthodox Christian | 0 (0%) | 0 (0%) | 868 (66%) | N/A |
| | Protestant/other Christian | 1,002 (43%) | 3,596 (60%) | 106 (8%) | 1,554 (44%) |
| | Muslim | 918 (39%) | 2,274 (38%) | 205 (25%) | 581 (17%) |
| | Traditional | 14 (1%) | 23 (0%) | 18 (1%) | N/A |
| | No religion | 2 (0%) | 0 (0%) | 1 (0%) | 10 (0%) |
| | Don't know | 0 (0%) | 2 (0%) | 0 (0%) | 0 (0%) |
| **Wealth quintile** | Lowest quintile | 293 (13%) | 2 (0%) | 321 (33%) | 329 (13%) |
| | 2nd quintile | 460 (20%) | 16 (0%) | 144 (13%) | 596 (23%) |
| | 3rd quintile | 370 (16%) | 147 (3%) | 147 (13%) | 354 (14%) |
| | 4th quintile | 477 (21%) | 940 (16%) | 197 (16%) | 624 (24%) |
| | Highest quintile | 656 (29%) | 4,738 (81%) | 351 (26%) | 693 (27%) |

[a] Data are n (%) or mean (SD)

N/A, not applicable

adjustment, we estimated a 9% absolute decrease in mCPR over time (95% CI: -17 to -0·3; p = 0·043; Table 3). The decrease in mCPR was driven by a decline in self-reported male condom use from 34% (95% CI: 31–37) at baseline to 19% (95% CI: 16–22) at endline. There was an 8% decrease in mCPR over time (95% CI: -15 to -0·3; p = 0·042) among unmarried girls and a 4% increase in married girls (95% CI: -6 to +14; p = 0·407).

This decrease in mCPR was accompanied by declines in several secondary outcomes. For instance, there was a 5% decrease (95% CI: -9 to -3; p<0·001) in the proportion of adolescent girls agreeing that contraception can help adolescent woman/girl to complete their education, find a better job and have a better life. There was also a 14% decrease (95% CI: -24 to -4; p = 0·008) on the intention to use a modern method, and a 9% increase (95% CI: 2–16; p = 0.019) in the proportion of modern contraceptive users using a LARC over time (S3 Table).

Self-reported exposure to Kuwa Mjanja was 24% (95% CI: 21–26). There was a positive association between self-reported exposure and mCPR (OR: 1·63; 95% CI: 1·28–2·09; p<0·001), and between self-reported exposure and five out of 17 secondary outcomes (S4 Table).

## Analysis of trend using other sources of data

In S1 Text, we present the results of the analysis of trends in modern contraceptive use for all three sites, using secondary data sources. In Nigeria, the data provided some evidence on the validity of the parallel trend assumption in Nasarawa and only weak evidence in Ogun. In

**Table 3. Association between mCPR (primary outcome) and Adolescents 360, and between mCPR and self-reported exposure, by site.**

| | | Nasarawa, Nigeria | Ogun, Nigeria | Oromia, Ethiopia | Mwanza, Tanzania |
|---|---|---|---|---|---|
| **Comparison of mCPR before and after A360** | mCPR in intervention areas at baseline [1] | 210/1,280 (16%) | 346/763 (45%) | 559/846 (61%) | 385/758 (51%) |
| | mCPR in intervention areas at endline [1] | 537/1,404 (38%) | 360/738 (49%) | 565/854 (61%) | 505/1,215 (42%) |
| | mCPR in comparison areas at baseline | 180/1,336 (13%) | 485/959 (51%) | N/A | N/A |
| | mCPR in comparison areas at endline | 422/1,560 (27%) | 413/810 (51%) | N/A | N/A |
| | Effect estimate (RR or RD) [2] | 0·96 (0·76 to 1·21) | 1·08 (0·92 to 1·26) | 0·05 (0·01 to 0·10) | -0·09 (-0·17 to -0·003) |
| | p-value | 0·738 | 0·340 | 0·025 | 0·043 |
| **Individual-level analysis** | mCPR among girls who reported being exposed to A360 in intervention areas | 50/98 (51%) | 22/54 (41%) | 177/217 (80%) | 167/309 (54%) |
| | mCPR among girls who reported not being exposed to A360 in intervention areas | 487/1326 (37%) | 338/684 (49%) | 388/637 (55%) | 338/906 (37%) |
| | Effect estimate (RR or OR) [3] | 1·43 (1·15 to 1·79) | 0·82 (0·59 to 1·14) | 2·09 (1·32 to 3·29) | 1·63 (1·28 to 2·09) |
| | p-value | 0·002 | 0·234 | 0·002 | <0·001 |

A360, adolescents 360 approach, mPCR, proportion of fecund and sexually active girls who reported using modern contraception at the time of the surveys, RR, risk ratio, RD, risk difference, OR, odds ratio

[1]This is a girl-level calculation

[2]In Nasarawa and Ogun, these correspond to adjusted risk ratios (RR) resulting from difference in difference analysis (Poisson regression) at girl-level; In Oromia and Mwanza, these are adjusted risk differences resulting from a pre-post comparison (linear regression) at kebele-level in Oromia and at street-level in Mwanza

[3]In Nasarawa and Ogun, these correspond to adjusted risk ratios (RR) resulting from Poisson regression models at girl-level; In Oromia and Mwanza, these are odds ratios (OR) resulting from logistic regression models at girl-level

Ethiopia, no clear trend was observed in mCPR, but in Tanzania, we observed an upward trend (S1 Text).

## Cost effectiveness

A360 design costs were seven to nine times higher than the comparator DELTA approach. A360 implementation costs were also substantially higher than maintaining existing contraceptive programming available to adolescents in the A360 study geographies. Incremental cost for the study geographies was $484,900 for Nasarawa, $513,220 for Ogun, $970,667 for Oromia, and $120,479 for Mwanza. The annual per capita (total population) spending on A360 design and implementation was between $0.22 in Mwanza and $0.67 in Oromia. As a percent of total health spending per capita (using WHO estimates of per capita health spending, adjusted to 2020 USD) [19], spending on A360 represented between 0·3% in Ogun and 2·5% in Oromia. Spending per eligible girl (sexually active, fecund, married/unmarried as per geography) per year of implementation ranged between $13 in Mwanza to $102 in Nasarawa (Table 4).

The mCPR changes translated to 4·4 maternal disability-adjusted life years (DALY) averted in Nasarawa, 17·0 in Ogun, 31·5 averted in Oromia, and 4·7 in Mwanza. Despite the lack of change in Nasarawa and the declines in modern contraceptive prevalence in Mwanza seen in the outcome evaluation, DALY impacts were positive due to increases in the number of eligible adolescents that result in positive additional users over the life of the project.

Dividing incremental costs by incremental effectiveness, produced an incremental cost per DALY averted of $111,416 (53 times gross domestic product [GDP] per capita) in Nasarawa, $30,114 (14 times GDP per capita) in Ogun, $30,855 (33 times GDP per capita) in Oromia, and $25,579 (24 times GDP per capita) in Mwanza (Table 5).

**Table 4. Design costs for A360 and comparator, implementation cost for A360 and comparator, incremental design costs, incremental implementation costs, and total incremental cost and intensity of spending on combined A360 design and implementation.**

| | | | Nasarawa, Nigeria | Ogun, Nigeria | Oromia, Ethiopia | Mwanza, Tanzania |
|---|---|---|---|---|---|---|
| **A360 and Comparator Costs** | Design cost | A360 | $85,603 | $46,643 | $123,724 | $17,024 |
| | | Comparator | $12,342 | $6,725 | $16,040 | $1,860 |
| | Implementation cost | A360 | $423,000 | $550,679 | $964,987 | $233,234 |
| | | Comparator | $11,361 | $77,378 | $102,003 | $127,920 |
| | Incremental cost | Design | $73,262 | $39,919 | $107,684 | $15,164 |
| | | Implementation | $411,638 | $473,302 | $862,983 | $105,314 |
| | | Total | $484,900 | $513,220 | $970,667 | $120,479 |
| **Intensity of spending on combined A360 design and implementation** | A360 cost per year per capita | | $0·40 | $0·21 | $0·67 | $0·22 |
| | As % of total health spend per capita | | 0·5% | 0·3% | 2·5% | 0·5% |
| | A360 cost per eligible girl per year of implementation | | $102 | $43 | $74 | $13 |

## Discussion

Our evaluation suggested evidence that A360 did not lead to increased adolescent use of modern contraceptives at a population level, except in Oromia, Ethiopia. Self-reported exposure to A360 was low, ranging from 5–24%, and in three settings was positively associated with mCPR. We observed a positive intervention impact on some secondary outcomes linked to the A360 Theory of Change. External data sources suggested an upward trend in mCPR among women aged 15–49 years. A360 was not cost-effective, with design costs higher than the comparator design approach. The absolute amount spent in relationship to population size and overall health spending was substantial.

The outcome evaluation had many strengths including accounting for clustering when calculating required sample sizes, the collection of comparable data before and after intervention implementation, use of multiple data sources to track population level mCPR and, in Nasarawa and Ogun, collection of data from populations both exposed and not exposed to the intervention. Participants were representative of A360 target populations increasing internal and external validity of the study. Data on self-reported exposure to the A360 programmes at endline allowed examination of the association between individual-level engagement with A360 and modern contraception use.

An important limitation was the lack of comparison areas in Mwanza and Oromia as a change in mCPR may reflect a time trend rather than an intervention effect [17, 20–22]. Our

**Table 5. Cumulative additional users, incremental disability-adjusted life years (DALY) averted, cost per DALY averted, and cost per DALY averted as a multiple of gross domestic product per capita.**

| Geography | Cumulative additional users | Cumulative Incremental DALYs averted = add decimal point | Cost per DALY averted | Cost per DALY averted times GDP per capita |
|---|---|---|---|---|
| **Nasarawa, Nigeria** | 44 | 4·4 | $111,416 | 53 |
| **Ogun, Nigeria** | 263 | 17·0 | $30,114 | 14 |
| **Oromia, Ethiopia** | 1,218 | 31·5 | $30,855 | 33 |
| **Mwanza, Tanzania** | 146 | 4·7 | $25,579 | 24 |

DALY, disability-adjusted life years, GDP, gross domestic product

findings may not be generalizable to other areas of the countries where A360 was implemented. We relied on respondent self-reporting to measure modern contraceptive use, sexual activity and exposure to the programs; these behaviours may have been subject to reporting bias. The COVID-19 pandemic led to changes to endline survey procedures including the use of face masks and the administration of the second section of the questionnaire by phone in Nasarawa, Ogun and Oromia. In Nasarawa and Ogun, participants in intervention and comparison areas were not entirely comparable in terms of sociodemographic factors. While we adjusted for changes in sociodemographic factors there may be some residual confounding. Finally, the validity of the difference in difference approach used in Nasarawa and Ogun depends on the mCPR time trend being the same in both intervention and comparison areas [17]. The only data available to evaluate similarity in time trends were Health Management Information System data from female clients aged 15–49 years using modern contraceptives at health facilities between early 2016 and mid-2020. These data reflect a broader age range than was included in the survey population and provided only weak evidence on the validity of the parallel trend assumption.

The costing and cost effectiveness study used a consistent approach in all three countries, repeated measures, and drew much of its information from accounting systems. Limitations are detailed in S1 Text. One-way and multi-way cost sensitivity analysis addressed many of these limitations, producing plausible lower and upper ranges to total cost used in sensitivity analysis.

Human-centred design (HCD) was a key element of the A360 approach. As defined by Giacomin [23], HCD 'is based on the use of techniques which communicate, interact, empathize and stimulate the people involved, obtaining an understanding of their needs', and it incorporates several features of social, behavioural and community engagement interventions. Deploying HCD in combination with youth engagement, insights from different disciplines, and working adaptively, helped the interventions resonate with girls, communities and government stakeholders [7]. HCD helped A360 integrate aspirational content, which attracted girls to events, built government buy-in, and allowed the program to operate in the context of high levels of stigma. In Oromia, messaging built around couples' counselling and financial planning, resonated strongly with married couples. Change was enabled by the integration of Smart Start into the Ethiopian government's Health Extension Program with delivery by Health Extension Workers, who are known and trusted in communities [24]. Among married girls aged 15–19 years in Nasarawa and Oromia, A360 implementation was associated with a population-level change in girls' belief in the benefits of modern contraception and in the proportion of girls with positive attitudes toward modern contraception.

Sociocultural factors remain a barrier to adolescent contraceptive use in Nasarawa, Ogun and Mwanza. For instance, in Nasarawa, one of the main reasons participants said they were not using contraception was the desire to bear children [25], reflecting established social norms [26]. In Ogun and Mwanza, A360 appeared less effective among unmarried girls perhaps because of insufficient impact on the stigma associated with premarital sex [27–29]. Unmarried girls need youth-friendly service delivery with ensured confidentiality and provider discretion. In both Ogun and Mwanza the majority of unmarried girls reported obtaining contraceptives at settings other than local health facilities. Process evaluation suggested A360 was not designed or resourced to address social norms [7]. A360 instead used light-touch approaches to engage communities, including: parents' sessions in Mwanza and couples' counselling in Oromia; working with community leaders, local government officials and trusted community structures; and, developing messaging that tapped into existing community concerns. Other studies suggest that community-level interventions need to be intensive and sustained to have long term impacts on knowledge, attitudes, practices and behaviors [4, 30].

In Mwanza, we observed a population level decrease in mCPR among adolescent girls. This might be explained by three main factors. First, during Kuwa Mjanja implementation in 2018, the Tanzanian former president made negative statements about contraception [31], and advertisements on contraception were later banned [32]. This at one point led to a halt to A360 outreach activities. Second, we saw an increase in the number of adolescent girls residing in study areas (which led to changes in sampling strategy at endline; see S1 Text), and a higher level of education at endline, both of which may have affected results.

The endline surveys of our outcome evaluation were conducted in late 2020, approximately one year after the start of COVID-19 pandemic. In Nigeria, data from Performance Monitoring for Action 2020 [33, 34] and from Krubiner and colleagues [35] indicate that family planning service and product availability for females aged 15–49 years are unlikely to have been impaired due to COVID-19, but it is difficult to know if this was also true among adolescent girls aged 15–19 years in our study geographies. In a State of Northern Nigeria (Kano), women aged 15–24 years who changed their contraceptive use status were more likely to adopt (4%) than to discontinue a method (<1%) [33]. On the other hand, in a State of Southern Nigeria (Lagos), women aged 15–24 years who changed their contraceptive use status were more likely to discontinue (11%) than to adopt a method (5%) [34], but only 2% of women aged 15–49 years stopped or interrupted their contraceptive method use due to COVID-19 restrictions [36]. In Ethiopia, even though some studies showed limited availability of family planning services and products due to COVID-19 [37–39], endline survey data, A360 monitoring data (collected by the implementers) and Health Management Information System service data, indicated that the effects of the pandemic were minimal [40]. Finally, in Tanzania, the COVID-19 pandemic stopped A360 service delivery between March and May 2020 [7], after which Kuwa Mjanja implementation was limited to door-to-door visits and short discussions, to avoid mass gatherings [7]. Nevertheless, the effects of COVID-19 on the supply of contraceptive commodities also seem limited in Tanzania, with the country following the World Health Organization guidance during the pandemic, by relaxing contraceptive prescription requirements and recommending that emergency contraception be available at pharmacies [35].

The low levels of reported exposure to A360 suggests that implementation intensity may have been lower than anticipated in the evaluation geographies and insufficient to achieve population-level change in mCPR. When the evaluation study was designed, the intervention implementation plans had not yet been finalised but the intention was for A360 to be implemented widely in the selected study geographies. As noted previously, HCD-based initiatives may require a phased evaluation approach with an outcome evaluation designed only when the programme and implementation strategy have been finalised [41]. Other alternative explanations for the low levels of reported exposure to A360 are: the tool was inappropriate to capture exposure to the program, although this is unlikely as the tool was validated with the implementers; high levels of migration, although this is also unlikely as in a secondary analysis we found very small proportion of girls having migrated for more than three months in the 12 months previous to the surveys.

The more intensive design effort in A360 was not cost effective in relation to the size of health outcomes achieved. Incremental cost-effectiveness ratios were far above the three times per capita GDP threshold for a cost-effective health intervention, per WHO-CHOICE standards [42]. They were also much higher than the $225 per DALY averted proposed as a cut-off for inclusion of interventions in Universal Health Care package, and far above the cost per DALY averted reported for other family planning interventions (between $235 and $587) [43]. Further analysis suggested that in Oromia, no level of mCPR increase would have led to cost-effective results, while in Nasarawa and Ogun large increases in mCPR would have been needed to reach a benchmark of three times GPD per capita. Efforts to reduce implementation

costs will be needed to produce more cost-effective models. Ethiopia and Nigeria are shifting management and service delivery responsibilities for the A360 legacy interventions to governments, which may lower costs. In Mwanza, A360 costs were more in line with potential impact, and even just maintaining baseline mCPR could have led to cost-effective results.

Our results highlight the challenges associated with identifying cost-effective approaches to increase the voluntary use of modern contraceptives among adolescents. A360 is being scaled-up and modified in Ethiopia and Nigeria, and expanded to Kenya [44]. According to the A360 implementers (PSI), the decision to scale-up was driven largely by the governments of the implementing countries. We recommend increased efforts to address social norms that prevent girls from accessing or using contraceptives as well as additional evaluation of A360 among both married and unmarried adolescents. Accordingly, A360's second phase aims to have a more consistent and rigorous approach to improving the enabling environment [44]. The modified A360 approaches are going to be evaluated through new cost-effectiveness studies.

## Supporting information

**S1 Text. Supporting information for the Adolescents 360 program evaluation.** S1 Text includes: a series of result figures; a detailed description of the methodology of the Adolescents 360 outcome evaluation and of the cost-effectiveness study; methods and results for the analysis of trends in modern contraceptive use prevalence; and the analysis plan of the Adolescents 360 outcome evaluation.
(DOCX)

**S2 Text. Questionnaire on inclusivity in global research.** S2 Text includes a questionnaire on inclusivity in global research outlining ethical, cultural, and scientific considerations that were taken into account during Adolescents 360 evaluations.
(DOCX)

**S3 Text. STROBE statement.** S3 Text includes a checklist of items that should be included in reports of observational studies.
(DOCX)

**S1 Table. Description of secondary outcomes, by Adolescents 360 theory of change components.** [1] Sexually active girls are those who report having sexual intercourse in the last 12 months. S1 Table presents a description of the secondary outcomes measured for the Adolescents 360 outcome evaluation, aligning with Adolescents 360 Theory of Change components.
(DOCX)

**S2 Table. Demographic characteristics of adolescent girls pre- and post-intervention, by site.** Data are n (%) or mean (SE). [1] p values are for differences between intervention and comparison areas pre-intervention. S2 Table presents a description of demographic characteristics of adolescent girls included in the Adolescents 360 outcome evaluation, pre- and post-intervention, by site.
(DOCX)

**S3 Table. Pre- versus post-intervention comparison of primary and secondary outcomes, by site.** mCPR, modern contraceptive prevalence rate, LARC, long-acting reversible contraceptive, Data are n (%) or mean (SE). [1] Girls who agreed with the sentence 'Using modern contraception can allow an adolescent woman girl to complete her education, find a better job and have a better life' [2] The impact of the Adolescents 360 approach is defined as the risk of mCPR pre- versus post-intervention. S3 Table presents a description of primary and secondary

outcomes measured for the Adolescents 360 outcome evaluation, pre- and post-intervention, by site. This table also presents the impact of the Adolescents 360 approach in each site.
(DOCX)

**S4 Table. Exposed versus non-exposed comparison of primary and secondary outcomes at endline, by site.** A360, Adolescents 360 approach, mCPR, modern contraceptive prevalence rate, LARC, long-acting reversible contraceptive, Data are n (%) or mean (SE). [1] Girls who agreed with the sentence 'Using modern contraception can allow an adolescent woman girl to complete her education, find a better job and have a better life' [2] Girls who agreed with the sentence 'Using modern contraception can allow a girl to achieve her life goals' [3] The impact of the A360 exposure is defined as the risk of mCPR in the exposed compared to girls not exposed to A360. S4 Table presents a description of primary and secondary outcomes measured for the Adolescents 360 outcome evaluation, by girls who reported being exposed to the intervention and girls who did not report being exposed to the intervention, by site.
(DOCX)

## Acknowledgments

We would like to thank all the interviewees, particularly the adolescent girls who engaged with us for sharing their perspectives. We thank Itad as the lead organization responsible for the overall A360 evaluation. We thank PSI Headquarters, PSI Ethiopia, PSI Tanzania and Society for Family Health for their engagement in conversations around the A360 programme and its implementation, and the evaluation study design; and their contributions to the cost study, and their participation in discussions on interpretation of the findings. We thank all the government officials, health workers, and community volunteers interviewed for the cost study.

## Author Contributions

**Conceptualization:** Catarina Krug, Melissa Neuman, James E. Rosen, Michelle Weinberger, Annapoorna Prakash, Mussa Kelvin Nsanya, Saidi Kapiga, Yewande P. Ajayi, Emily E. Crawford, Eskindir Tenaw, Mohammed Mussa, Christian Bottomley, James R. Hargreaves, Aoife Margaret Doyle.

**Data curation:** Catarina Krug, James E. Rosen, Michelle Weinberger, Annapoorna Prakash, Mussa Kelvin Nsanya, Philip Ayieko, Yewande P. Ajayi, Emily E. Crawford, Eskindir Tenaw, Som Kumar Shrestha.

**Formal analysis:** Catarina Krug, James E. Rosen, Michelle Weinberger, Annapoorna Prakash.

**Funding acquisition:** Melissa Neuman, Stefanie Wallach, Mary Lagaay, Melanie Punton, Aoife Margaret Doyle.

**Investigation:** Mussa Kelvin Nsanya, Philip Ayieko, Saidi Kapiga, Yewande P. Ajayi, Emily E. Crawford, Eskindir Tenaw, Mohammed Mussa.

**Methodology:** Catarina Krug, James E. Rosen, Michelle Weinberger, Annapoorna Prakash, Mussa Kelvin Nsanya, Philip Ayieko, Yewande P. Ajayi, Emily E. Crawford, Eskindir Tenaw, Som Kumar Shrestha, Christian Bottomley, James R. Hargreaves, Aoife Margaret Doyle.

**Project administration:** Catarina Krug, Mussa Kelvin Nsanya, Philip Ayieko, Saidi Kapiga, Yewande P. Ajayi, Emily E. Crawford, Eskindir Tenaw, Mohammed Mussa, Aoife Margaret Doyle.

**Resources:** Catarina Krug, James E. Rosen, Michelle Weinberger, Annapoorna Prakash.

**Software:** Catarina Krug, James E. Rosen, Michelle Weinberger, Annapoorna Prakash.

**Supervision:** Catarina Krug, Melissa Neuman, Mussa Kelvin Nsanya, Philip Ayieko, Saidi Kapiga, Yewande P. Ajayi, Emily E. Crawford, Eskindir Tenaw, Mohammed Mussa, Aoife Margaret Doyle.

**Validation:** Melissa Neuman, Annapoorna Prakash, Christian Bottomley, James R. Hargreaves, Aoife Margaret Doyle.

**Visualization:** Catarina Krug, James E. Rosen, Michelle Weinberger.

**Writing – original draft:** Catarina Krug.

**Writing – review & editing:** Catarina Krug, Melissa Neuman, James E. Rosen, Michelle Weinberger, Stefanie Wallach, Mary Lagaay, Melanie Punton, Annapoorna Prakash, Philip Ayieko, Saidi Kapiga, Yewande P. Ajayi, Emily E. Crawford, Eskindir Tenaw, Mohammed Mussa, Som Kumar Shrestha, Christian Bottomley, James R. Hargreaves, Aoife Margaret Doyle.

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
