## [Decision Letter · Decision Letter 0]

6 Jun 2023

PGPH-D-23-00474

Effect and cost-effectiveness of human-centred design-based approaches to increase adolescent uptake of modern contraceptives in Nigeria, Ethiopia and Tanzania: population-based, quasi-experimental studies

Dear Dr. Krug,

Thank you for submitting your manuscript to PLOS Global Public Health. After careful consideration, we feel that it has merit but does not fully meet PLOS Global Public Health’s publication criteria as it currently stands. Therefore, we invite you to submit a revised version of the manuscript that addresses the points raised during the review process.

Two reviewers have provided constructive comments. Please carefully consider all feedback, particularly requests for clearer descriptions of interventions implemented and concerns about adequacy of analytical methods for stated causal inferences. 

We look forward to receiving your revised manuscript.

Kind regards,

Hannah Tappis, DrPH, MPH

Academic Editor

Journal Requirements:

Additional Editor Comments (if provided):

If resubmitting a revised manuscript, please ensure text adheres to reporting standards for implementation studies, observational studies and cost-effectiveness evaluation, available at https://www.equator-network.org/.

Reviewers' comments:

Reviewer's Responses to Questions

**Comments to the Author**

1. Does this manuscript meet PLOS Global Public Health’s publication criteria? Is the manuscript technically sound, and do the data support the conclusions? The manuscript must describe methodologically and ethically rigorous research with conclusions that are appropriately drawn based on the data presented.

Reviewer #1: No

Reviewer #2: Yes

2. Has the statistical analysis been performed appropriately and rigorously?

Reviewer #1: No

Reviewer #2: Yes

3. Have the authors made all data underlying the findings in their manuscript fully available (please refer to the Data Availability Statement at the start of the manuscript PDF file)?

Reviewer #1: Yes

Reviewer #2: Yes

4. Is the manuscript presented in an intelligible fashion and written in standard English?

Reviewer #1: Yes

Reviewer #2: Yes

5. Review Comments to the Author

Reviewer #1: See report attached.

Reviewer #2: This is an impressive and enormous project and your evaluation accordingly.

My suggestions to make the manuscript easier to read for those not as familiar with your acronyms and methods are detailed below.

Intro or methods

1. Please consider giving a more detailed account of how Covid19 affected your study areas and potentially the impact of interventions in the study other than survey methods. Consider this in your discussion.

2. P.5. Lines 100- It is to your considerable credit that you included a design effect in your sampling to take account of cluster, thus increasing the sizes you needed to reach. Many studies don't. Why not state this.

3. Please explain that while 'modern contraception included many effective and not so effective methods, that you included a measure of LARC in your outcomes. It is important but is not clear to the reader if they don't read all your supplements.

Results

4. You appeared to me to achieve your sample size in three out of four areas. I think you should make a more comprehensible description of your actual samples. I found Figure 1 complex and difficult to understand what you meant by base and endline samples that look as if you are fine, but below you state the actual number interviewed and some are quite small.

5. Please label the purpose of each of your Supplementary files at the top please.

Discussion

6. P 21 line 337 - explain again what HCD is please.

7. Ogun sample is very wealthy compared to much of the other areas. Would this have had any effect on the outcome?

Congratulations on a logistically very complex but important study.

6. PLOS authors have the option to publish the peer review history of their article (what does this mean?). If published, this will include your full peer review and any attached files.

**Do you want your identity to be public for this peer review?** For information about this choice, including consent withdrawal, please see our Privacy Policy.

Reviewer #1: No

Reviewer #2: **Yes: **Angela Taft

---

## [Editor Report · Decision Letter 1]

29 Aug 2023

PGPH-D-23-00474R1

Effect and cost-effectiveness of human-centred design-based approaches to increase adolescent uptake of modern contraceptives in Nigeria, Ethiopia and Tanzania: population-based, quasi-experimental studies

Dear Dr. Krug,

Thank you for submitting your manuscript to PLOS Global Public Health, and thoughtfully addressing previous reviewer comments. After careful consideration, we feel that the manuscript meets most publication criteria but would benefit from additional reflection on a few points noted in the editorial comments below. Therefore, we invite you to submit a revised version of the manuscript that addresses these points raised during the review process.

We look forward to receiving your revised manuscript.

Kind regards,

Hannah Tappis, DrPH, MPH

Academic Editor

Journal Requirements:

Additional Editor Comments (if provided):

This is an important evaluation of a flagship adolescent reproductive health program. Both the findings and reflections on methodological limitations encountered will be of great interest to health practitioners and policy makers engaged in or considering similar programs in other settings. In the spirit of open science, the inclusion of a detailed annex including intervention characteristics and documentation of analyses conducted is also appreciated and will serve as an example for other researchers.

It is interesting to note that A360 approaches are being scaled up in Ethiopia and Nigeria, and expanded to Kenya, despite evidence that it is not a cost-effective intervention package and is not designed to address social norms that remain a barrier to adolescent contraceptive use across program locations. I was somewhat surprised to read that the program is being expanded, and feel it merits further attention in the Discussion. Can the authors provide additional insights into the decision to scale-up? Were modifications to the intervention approach recommended based on evaluation results, and are these being adopted in expansion sites?
---

## [Editor Report · Decision Letter 2]

27 Sep 2023

Effect and cost-effectiveness of human-centred design-based approaches to increase adolescent uptake of modern contraceptives in Nigeria, Ethiopia and Tanzania: population-based, quasi-experimental studies

PGPH-D-23-00474R2

Dear Dr Krug,

We are pleased to inform you that your manuscript 'Effect and cost-effectiveness of human-centred design-based approaches to increase adolescent uptake of modern contraceptives in Nigeria, Ethiopia and Tanzania: population-based, quasi-experimental studies' has been provisionally accepted for publication in PLOS Global Public Health.

Best regards,

Hannah Tappis, DrPH, MPH

Academic Editor